# Interplay between MPIase, YidC, and PMF during Sec-independent insertion of membrane proteins

Yuta Endo[1], Yuko Shimizu[2], Hanako Nishikawa[2], Katsuhiro Sawasato[2], Ken-ichi Nishiyama[1,2]

Integral membrane proteins with the N-out topology are inserted into membranes usually in YidC- and PMF-dependent manners. The molecular basis of the various dependencies on insertion factors is not fully understood. A model protein, Pf3-Lep, is inserted independently of both YidC and PMF, whereas the V15D mutant requires both YidC and PMF in vivo. We analyzed the mechanisms that determine the insertion factor dependency in vitro. Glycolipid MPIase was required for insertion of both proteins because MPIase depletion caused a significant defect in insertion. On the other hand, YidC depletion and PMF dissipation had no effects on Pf3-Lep insertion, whereas V15D insertion was reduced. We reconstituted (proteo)liposomes containing MPIase, YidC, and/or $F_0F_1$-ATPase. MPIase was essential for insertion of both proteins. YidC and PMF stimulated Pf3-Lep insertion as the synthesis level increased. V15D insertion was stimulated by both YidC and PMF irrespective of the synthesis level. These results indicate that charges in the N-terminal region and the synthesis level are the determinants of YidC and PMF dependencies with the interplay between MPIase, YidC, and PMF.

## Introduction

Membrane proteins co-translationally are inserted into the cytoplasmic membranes of *Escherichia coli* with the aid of a series of insertion factors, such as signal recognition particle (SRP)/SRP receptor (SR), the SecYEG translocon and YidC (for reviews, see references 1, 2, and 3). In addition to these proteinaceous factors, glycolipid MPIase is involved in protein insertion (4, 5). As far as we know at present, MPIase cooperates with SecYEG and YidC to insert membrane proteins because MtlA insertion requires both MPIase and SecYEG (6, 7), and both MPIase and YidC are required for membrane insertion of the c subunit of $F_0F_1$-ATPase ($F_0$c) (8, 9). Beside SecYEG-dependent insertion, it is known that Sec-independent insertion occurs. In this case, it has been thought that membrane proteins are inserted into membranes spontaneously or unassisted through the

hydrophobic interaction between membrane lipids and the transmembrane (TM) domains of proteins because a subset of membrane proteins, such as M13 procoat, Pf3 coat and $F_0$c, are inserted into liposomes of phospholipids only (10, 11, 12). However, inclusion of DAG in liposomes at a physiological level blocks spontaneous insertion completely (6, 13, 14), indicating that spontaneous insertion does not occur in vivo. Alternatively, MPIase (15, 16) and YidC (17, 18) are involved in the insertion. We have observed a cooperative function of MPIase with YidC, in which MPIase functions at an early stage and then the substrate proteins are transferred to YidC to complete insertion (7, 8, 9).

Sec-independent insertion is still not fully understood. M13 procoat and Pf3 coat are inserted into membranes in YidC- and proton motive force (PMF)-dependent manners (19, 20), whereas the respective mutants, such as 3L-Pf3 coat, sometimes render insertion YidC-independent or PMF-independent or even independent of both (21, 22). $F_0$c insertion depends upon YidC but not PMF (23). Therefore, the mechanisms underlying these differences in dependencies on YidC and PMF are totally unknown. Although we have shown that MPIase is involved in the insertion of abovementioned proteins (6, 7, 8, 16), it remains unknown whether or not MPIase is generally involved in all the types of insertion described above, that is, it is unknown whether YidC/Sec-independent, YidC only, YidC/PMF only, or YidC/Sec mechanisms co-exist and operate in an MPIase-dependent manner. The insertion factor dependencies for proteins described above were summarized in Table 1. MPIase was essential for insertion all of them, whereas the dependencies on SecYEG, YidC, and PMF quite differ.

A model substrate protein, Pf3-Lep, is composed of the periplasmic region of Pf3 coat protein, followed by TM1 of Lep (leader peptidase) (24). Whereas Pf3 coat insertion is YidC-dependent (17, 22), Pf3-Lep insertion is not affected by YidC depletion (24). In the case of PMF, Pf3 coat insertion is significantly stimulated by PMF, but Pf3-Lep insertion is not affected by PMF dissipation (14, 24). On the other hand, the V15D mutant of Pf3-Lep is severely affected by both YidC depletion and PMF dissipation (24). These differences in insertion factor dependencies were also summarized in Table 1. Therefore, these model proteins are good substrates to analyze the switching of insertion dependency. In this study, we analyzed the molecular mechanisms underlying the membrane insertion of

[1]The United Graduate School of Agricultural Sciences, Iwate University, Morioka, Japan   [2]Department of Biological Chemistry and Food Science, Faculty of Agriculture, Iwate University, Morioka, Japan

Correspondence: nishiyam@iwate-u.ac.jp

**Table 1.  Insertion factor dependency for membrane proteins determined in the in vitro reconstitution system.**

| Protein | SecYEG | MPIase | YidC | Proton motive force | References |
|---|---|---|---|---|---|
| MtlA | Essential | Essential | Conditional | No effect | 6 and 7 |
| Pf3 coat | Not essential | Essential | Stimulate | Stimulate | 7 and 22 |
| 3L-Pf3 coat | Not essential | Essential | Conditional | No effect | 7, 15, 16, and 22 |
| M13 procoat | Not essential | Essential | Stimulate | Stimulate | 6, 7, and 14 |
| $F_0c$ | Not essential | Essential | Stimulate | No effect | 8, 9, and 23 |
| Pf3-Lep | Not essential | (This study) | Not required | No effect | 24 |
| Pf3-Lep V15D | Not essential | (This study) | Stimulate | Stimulate | 24 |

The insertion factor dependencies for specified membrane proteins are indicated. "Conditional" denotes that YidC stimulates insertion when the substrate level is high. In the case of Pf3-Lep and Pf3-Lep V15D, used in this study, the in vivo results are indicated.

these substrates to clarify the switching, by means of an in vitro system. We found that both substrates are inserted into membranes in an MPIase-dependent manner, and that charges in the N-terminal region and the synthesis level are determinants of the YidC and PMF dependencies with the interplay between MPIase, YidC, and PMF.

# Results

## In vitro analysis of Pf3-Lep (amber) and V15D insertion

Pf3-Lep (amber) is a model membrane protein, in which the periplasmic region of Pf3 coat protein is followed by the first TM and the cytoplasmic region of Lep with three stop codons after 79E (Fig 1A, left; Fig S1). An amino acid substitution of 15V with D was introduced into the V15D mutant (Fig 1A, right). To examine the dependency on insertion factors, MPIase-depleted (KS23) and YidC-depleted (JS7131) INV (inverted and inner membrane vesicles) were prepared (Fig 1B). The successful depletion of either MPIase or YidC was confirmed by immunoblotting (Fig 1B). Up-regulation of MPIase was also observed upon YidC depletion, as reported (7). The insertion activity was determined by means of a protease protection assay (Fig 1C) (25). The membrane insertion region is protected by proteinase K (PK), giving membrane-protected fragments (MPFs). At first, insertion of both proteins into the wild-type INV (EK413) was examined. Upon PK digestion, mainly three bands appeared (Fig 1D). Band "c" appeared only in the presence of INV, indicating that this band represents the MPF. Band "i" was a membrane-embedded fragment digested at the N terminus. Because the intensity of this band was very weak, we did not characterize this material further. Band "r" appeared even in the absence of INV, indicating that this band does not correspond to any insertion, but that it is a PK-resistant material derived from the hydrophobic TM domain. This band is indicated by an asterisk at the left of each gel. Hereafter, we focused on the appearance of the "c" band as an index of membrane insertion. When SRP/SR (Ffh/FtsY) was added to the reaction mixture, a significant increase in the insertion of both proteins was observed, indicating that the TM domain was recognized by SRP to solubilize substrates. Therefore, we added SRP/SR in the following experiments. When YidC-depleted INV were used, no inhibition of Pf3-Lep (amber) insertion was observed,

whereas a decrease in V15D insertion was observed (Fig 1E), consistent with in vivo results (24). On the other hand, when MPIase-depleted INV were used, a significant decrease in insertion of both substrates was observed, strongly suggesting that these substrate proteins require MPIase for insertion (Fig 1E).

Next, we examined whether SecA and PMF are involved in Pf3-Lep (amber) insertion (Fig 2). SecA activity is inhibited by sodium azide (26). When azide was added to the reaction mixture, pOmpA translocation was completely abolished (Fig 2A). Under our experimental conditions, PMF is generated by $F_0F_1$-ATPase, of which the activity is abolished by N,N-dicyclohexylcarbodiimide (DCCD) (27). Insertion of Pf3 coat protein is stimulated by the membrane potential, a component of PMF (10, 20). When DCCD was added to the reaction mixture, Pf3 coat insertion was significantly reduced (Fig 2B). Insertion of both Pf3-Lep (amber) and V15D was not affected by the addition of azide, indicating that SecA is not involved in the insertion of these proteins (Fig 2C). DCCD addition did not affect Pf3-Lep (amber) insertion, indicating that Pf3-Lep (amber) insertion is PMF-independent (Fig 2D, left). On the other hand, V15D insertion was reduced by ~2/3, indicating that PMF is involved in V15D insertion (Fig 2D, right). When ΔYidC INV were used, Pf3-Lep (amber) insertion was even affected by DCCD addition, indicating that efficient insertion of Pf3-Lep (amber) requires PMF in the absence of YidC (Fig 2E, left). In the case of V15D insertion, the insertion into ΔYidC INV was as low as that into WT INV in the absence of PMF, indicating that both YidC and PMF are involved in V15D insertion. In summary, Pf3-Lep (amber) insertion is independent of both YidC and PMF, but is reduced in the absence of both. On the other hand, V15D insertion requires both YidC and PMF for the efficient insertion in the in vitro assay system using INV. These results also suggest that the YidC function is tightly linked with PMF utilization for stimulation of membrane insertion.

## Reconstitution analysis reveals interplay between MPIase, YidC, and PMF

Next, we tried to reconstitute Pf3-Lep (amber) and V15D insertion (Fig 3). When liposomes composed of only phospholipids (PL) were used, efficient spontaneous insertion of both proteins, which does not reflect the in vivo reaction (6, 13), was observed (Fig 3A, "PL"). This disordered spontaneous insertion was blocked by the presence of DAG ("PL+DAG"). When YidC was included in PL+DAG

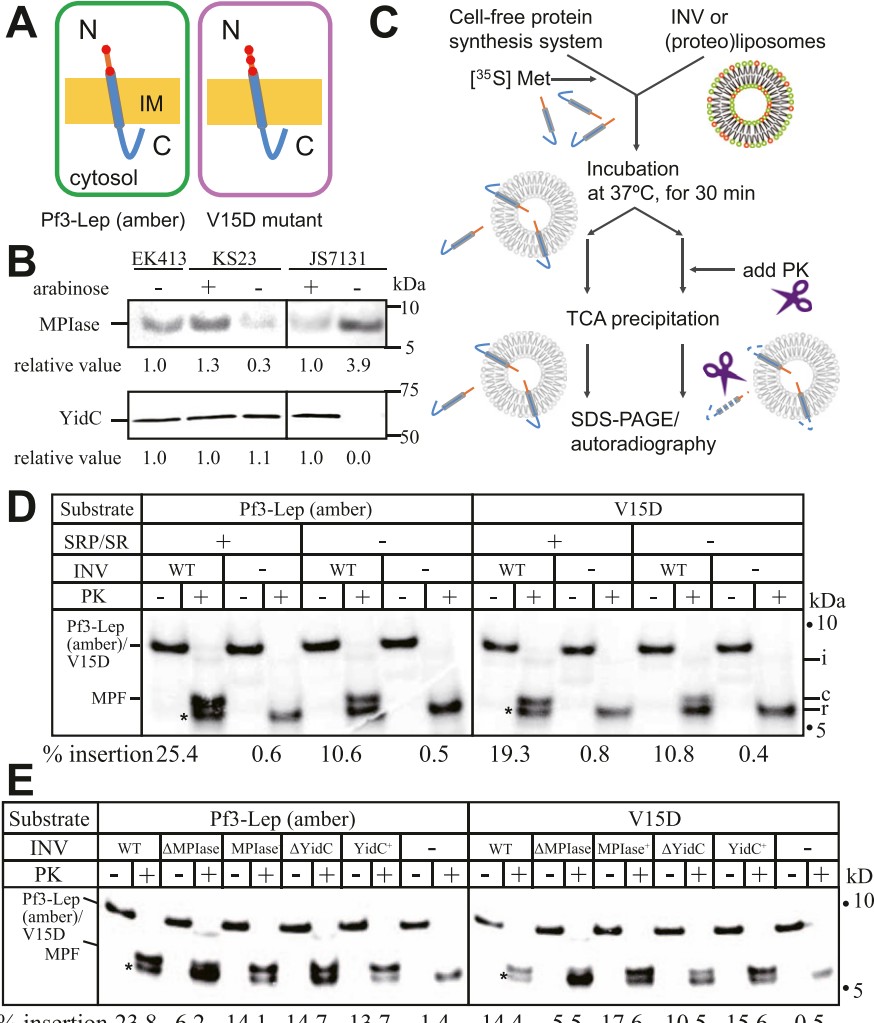

**Figure 1. MPIase is involved in membrane insertion of Pf3-Lep (amber).**
**(A)** Membrane topologies of Pf3-Lep (amber) (left) and its V15D mutant (right). The N-terminal region is exposed to the periplasm and the C-terminal region to the cytosol. Negatively charged residues in the periplasmic region are denoted by red dots. **(B)** Preparation of MPIase- and YidC-depleted INV. Depletion of MPIase (upper panel) and YidC (lower panel) was confirmed by immunoblotting. Note that MPIase is up-regulated by YidC depletion. The relative levels of MPIase and YidC are shown below the blots. Whole cell extract (1 μg protein) and INV (10 μg protein) were used to detect MPIase and YidC, respectively. **(C)** Schematic representation of assaying of membrane insertion in vitro. A membrane-protected fragment (MPF) arises upon PK digestion. The C-terminal region of Pf3-Lep (amber) is digested, giving the MPF. **(D)** Membrane insertion of both Pf3-Lep (amber) and V15D is stimulated by signal recognition particle. The substrates were in vitro synthesized in the presence of INV prepared from EK413 (WT) as specified. Upon PK digestion after synthesis, three bands, "i," "c," and "r," appeared. Band "i" represents incomplete insertion, while band "c" represents the MPF. Band "r" is the PK-resistant band because this non-specifically appeared even in the absence of membranes. Hereafter, this band is indicated by asterisks. The insertion activity (the percentage of the level of "c" as to that of the substrates) is shown below the autoradiograms. The numbers of methionine (three in the substrates and two in MPF) were considered in the calculation. The positions of molecular weight markers are also shown by dots. **(E)** Membrane insertion of Pf3-Lep (amber) and V15D into MPIase- and YidC-depleted INV. **(B)** The substrates were in vitro synthesized in the presence of ΔMPIase INV and ΔYidC INV, shown in (B), as specified. **(D)** The insertion activity was determined as described in (D) and is shown.
Source data are available for this figure.

liposomes, no insertion activity for either protein was observed, indicating that YidC is not sufficient for insertion (Fig 3B, left). On the other hand, both proteins were inserted into MPIase liposomes, albeit at a low level (basal level) (Fig 3B, middle), indicating that MPIase is essential for the insertion of both proteins. These activities reached a plateau level because an increase in the amount of MPIase had no effect on the activity (Fig 3B, right). When YidC was co-reconstituted with MPIase (+YidC), Pf3-Lep (amber) insertion was stimulated (Fig 3B, middle and right) to a level comparable with that into WT INV (Figs 1 and 2). On the other hand, V15D insertion was not significantly stimulated by YidC (Fig 3B, middle and right), unlike the case of WT INV (Figs 1 and 2). We assumed that YidC had little effect on V15D insertion because of the absence of PMF. We imposed PMF by co-reconstituting $F_0F_1$-ATPase together with MPIase and/or YidC (Fig 3C). When Pf3-Lep (amber) insertion was analyzed, the maximum activity was obtained in the presence of MPIase only, similarly to in the presence of both MPIase and YidC (Fig 3C, left). Although the activity of V15D insertion was as low as that in the absence of YidC, it was clearly stimulated to the level into WT INV in the presence of both MPIase and YidC (Fig 3C, right). Thus, the results in the reconstitution experiments reproduced those using

INV, that is, we found that a sufficient level of Pf3-Lep (amber) insertion is obtained when either MPIase and YidC or MPIase and PMF are present. On the other hand, all three factors, MPIase, YidC, and PMF, are necessary for a sufficient level of V15D insertion.

We also found that the expression level of the substrate proteins is an important parameter that determines the insertion efficiency (Fig 4). When the expression level was increased by increasing the amounts of radioactive methionine (~2 MBq/ml in Figs 1–3, and ~10 MBq/ml in Fig 4), the expression level increased from 0.34~0.71 pmol/ml (Figs 1–3) to 4.1~4.9 pmol/ml (Fig 4A). In this case, Pf3-Lep (amber) insertion into MPIase/YidC proteoliposomes in the absence of PMF was as efficient as in the presence of PMF, however, insertion into MPIase liposomes remained low even in the presence of PMF (upper panel), unlike as shown in Fig 3C, indicating that an increase in the substrate amounts rendered Pf3-Lep (amber) insertion YidC-dependent. The maximum activity of V15D insertion was obtained only when the three factors, MPIase, YidC and PMF, were present (lower panel). Whereas V15D insertion was observed when MPIase was present, YidC addition only or PMF imposition only had little effect on V15D insertion. When cold methionine was added to the reaction mixture, the synthesis level was further

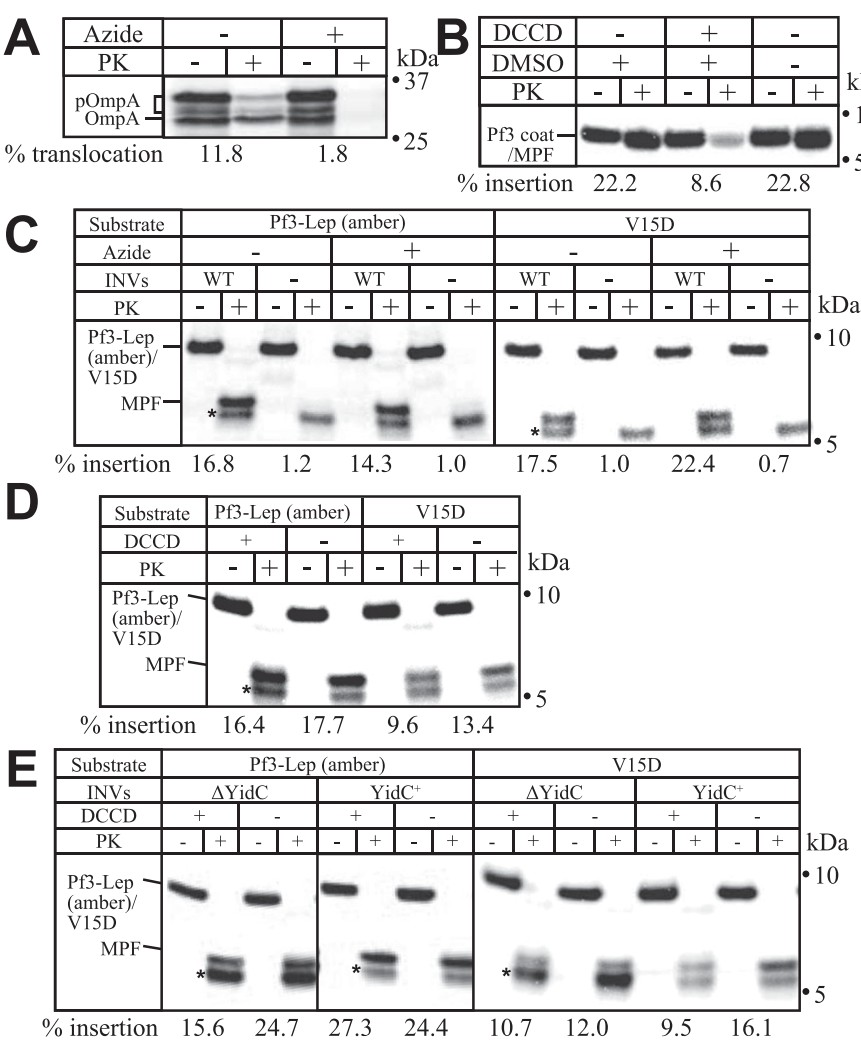

**Figure 2. Effects of SecA and proton motive force (PMF) on Pf3-Lep (amber) and V15D insertion.**
**(A)** Effect of sodium azide on SecA-dependent pOmpA translocation. pOmpA was in vitro synthesized in the presence of INV prepared from EK413, followed by the PK protection assay. Sodium azide (1 mM) was added as specified. The percentage of the translocated materials (pOmpA plus OmpA) is shown at the bottom. The numbers of methionine (six in pOmpA and five in OmpA) were considered in the calculation. **(B)** Effect of DCCD on PMF-dependent stimulation of Pf3 coat insertion. Pf3 coat was in vitro synthesized in the presence of INV prepared from EK413, followed by the PK protection assay. DCCD (0.15 mM) or DMSO was added as specified. The insertion activity was determined and is shown at the bottom. **(C)** Effect of sodium azide on Pf3-Lep (amber)/V15D insertion. The insertion activity for Pf3-Lep (amber) (left half) and V15D (right half) was determined as described in the legend to Fig 1C, and is shown at the bottom. **(A)** Where specified, sodium azide was added as in (A). The position of membrane-protected fragment is indicated. The PK-resistant bands that appeared in the absence of INV as indicated by asterisks. **(D, E)** Effect of DCCD on Pf3-Lep (amber)/V15D insertion. The insertion activity for Pf3-Lep (amber) (left half) and V15D (right half) was determined as described in the legend to Fig 1D, and is shown at the bottom. **(B)** DCCD was added as in (B). **(D, E)** INV prepared from EK413 were used in (D), whereas ΔYidC or YidC[+] INV prepared from JS7131 were used in (E) as specified. The position of membrane-protected fragment is indicated. The PK-resistant bands unrelated with membrane insertion are indicated by asterisks.

increased to 390~530 pmol/ml (Fig 4B). In this case, the presence of both MPIase and YidC did not lead to the maximum activity of Pf3-Lep (amber) insertion found in the presence of the three factors (upper panel), unlike as shown in Fig 4A. Similarly, V15D insertion required the three factors (lower panel). These results indicate that the MPIase-dependent insertion of both substrate proteins becomes dependent on both YidC and PMF as the expression level of the substrates increased.

## MPIase and YidC interact directly

The results so far obtained reveal the presence of functional interaction between MPIase and YidC. Therefore, we examined whether or not MPIase and YidC directly interact by means of the pull-down assay. JS7131, a *yidC*-disrupted strain, expresses YidC at the wild-type level in the presence of arabinose (17). When plasmid pTac-YidC-CHis was used to transform in JS7131, the transformants grew very well in the absence of inducer IPTG with leaky expression of His-tagged YidC at the wild-type level

(Fig 5A). INV were prepared from both strains, followed by the pull-down assay. When solubilized membranes were subjected to cobalt metal-affinity column chromatography, His-tagged YidC was purified from sample of JS7131/pTac-YidC-CHis, but not from that of JS7131, as expected (Fig 5B). The same fraction was also analyzed to detect MPIase, by immunostaining of the TLC plates. Fig 5C clearly shows that the fraction with His-tagged YidC contains MPIase, indicating that YidC and MPIase were co-precipitated. These results strongly suggest that MPIase and YidC interact directly in the membranes.

## Discussion

In this study, we examined the molecular mechanism underlying the insertion of N-out membrane proteins using Pf3-Lep (amber) and its mutant V15D as substrates by means of INV and reconstituted (proteo)liposomes. All the results indicated that glyco-lipid MPIase is essential for the insertion of both proteins. Under

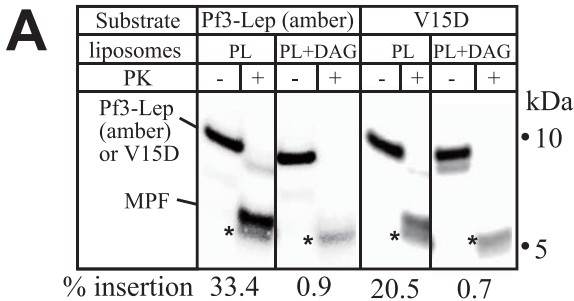

**A**

| Substrate | Pf3-Lep (amber) | | V15D | |
|---|---|---|---|---|
| liposomes | PL | PL+DAG | PL | PL+DAG |
| PK | - + | - + | - + | - + |

Pf3-Lep (amber) or V15D

MPF

\*      \*      \*      \*

kDa
• 10
• 5

| % insertion | 33.4 | 0.9 | 20.5 | 0.7 |

**B**

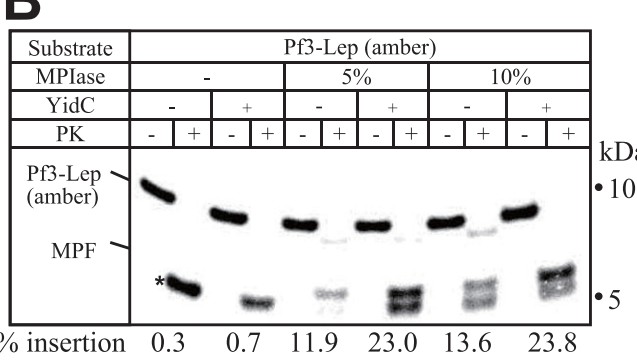

| Substrate | Pf3-Lep (amber) | | | | | |
|---|---|---|---|---|---|---|
| MPIase | - | | 5% | | 10% | |
| YidC | - | + | - | + | - | + |
| PK | - + | - + | - + | - + | - + | - + |

Pf3-Lep (amber)

MPF

\*

kDa
• 10
• 5

| % insertion | 0.3 | 0.7 | 11.9 | 23.0 | 13.6 | 23.8 |

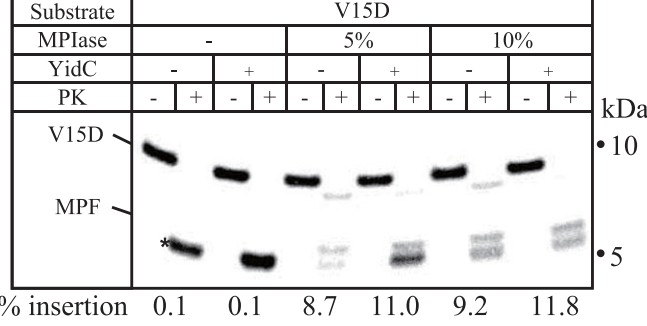

| Substrate | V15D | | | | | |
|---|---|---|---|---|---|---|
| MPIase | - | | 5% | | 10% | |
| YidC | - | + | - | + | - | + |
| PK | - + | - + | - + | - + | - + | - + |

V15D

MPF

\*

kDa
• 10
• 5

| % insertion | 0.1 | 0.1 | 8.7 | 11.0 | 9.2 | 11.8 |

**C**

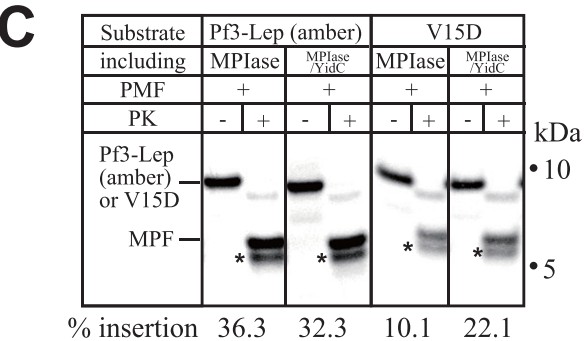

| Substrate including | Pf3-Lep (amber) | | V15D | |
|---|---|---|---|---|
| | MPIase | MPIase/YidC | MPIase | MPIase/YidC |
| PMF | + | + | + | + |
| PK | - + | - + | - + | - + |

Pf3-Lep (amber) or V15D

MPF

\*      \*      \*      \*

kDa
• 10
• 5

| % insertion | 36.3 | 32.3 | 10.1 | 22.1 |

**Figure 3.  Reconstitution of Pf3-Lep (amber) and V15D insertion.**
**(A)** Spontaneous insertion of both Pf3-Lep (amber) and V15D is blocked by a physiological level of DAG. Pf3-Lep (amber) (left half) and V15D (right half) were in vitro synthesized in the presence of liposomes formed with phospholipids (PL) or phospholipids and DAG (PL+DAG), followed by the PK protection assay. The insertion activity was determined and is shown at the bottom. The position of membrane-protected fragment is indicated. The PK-resistant bands unrelated with membrane insertion are indicated by asterisks. **(B)** Effects of MPIase and YidC on Pf3-Lep (amber) and V15D insertion. (Proteo)liposomes containing MPIase and YidC were reconstituted, followed by assaying of Pf3-Lep (amber) (upper panel) and V15D (lower panel) insertion. The insertion activity was determined as described in the legend to Fig 1D and is shown at the bottom.

all the conditions examined, both Pf3-Lep (amber) and the V15D mutant were inserted into MPIase liposomes, albeit to a basal level. To achieve efficient insertion, YidC and PMF are required in addition. The importance of YidC and PMF was determined by the presence of the charge at the N-terminal domain and the expression level. The presence of the charge (V15D in this case) and the higher expression level of the substrates render the insertion more dependent on YidC and PMF. Thus, the interplay of the three factors, MPIase, YidC, and PMF, causes efficient insertion.

Interplay between MPIase and YidC is supported by the fact that YidC depletion causes MPIase upregulation (7), suggesting the occurrence of functional interaction between MPIase and YidC. Pull-down assay actually supported the direct interaction between MPIase and YidC. The early stage of insertion is initiated by MPIase, followed by the YidC function to complete insertion at the late stage (7, 8). Because PMF was not imposed in these studies, interaction of substrate proteins with MPIase and YidC occur even in the absence of PMF, as seen in this study as well. The YidC function is powered by PMF, as indicated by the crystal structure of YidC (18, 28). The positive charge in the cavity in the membrane-embedded domain of YidC interacts with the negative charge found in the N-terminal region of the membrane protein through an electrostatic interaction. Then, the negative charge is translocated to the periplasmic space to complete insertion. This step would be accelerated with the aid of PMF (18). This model also explains that PMF stimulates insertion more effectively in the presence of YidC. As a result, MPIase and YidC could function for the next insertion reaction. This model implies that YidC and PMF become less important for insertion when the expression level of the substrates is lower, or the number of negative charges in the N-terminal region decreases (Fig 6). In an extreme case, only MPIase is even sufficient for insertion (Fig 6, left box). The insertion of some membrane proteins, such as 3L-Pf3 coat (10), is independent of both YidC and PMF (22), similar to in the case for Pf3-Lep (24), suggesting that the YidC-independent mechanism is operative. In this case, MPIase should be required as shown in this study. On the other hand, we have shown that 3L-Pf3 coat insertion becomes YidC-dependent when the expression level increases (7). Thus, interplay between MPIase, YidC, and PMF is important for the membrane insertion of N-out membrane proteins, whereas the number of charges at the N terminus and the expression level affect the dependency on YidC and PMF.

In the previous report (24), it is concluded that introduction of both negatively and positively charged residues in the N-terminal region of Pf3-Lep renders insertion YidC-dependent. Both net charge numbers and charge distribution are important to determine the YidC-dependence. In this regard, our reconstitution

**(C)** Effects of PMF on Pf3-Lep (amber) and V15D insertion. $F_0F_1$-ATPase was co-reconstituted with MPIase and YidC to impose PMF. The proteoliposomes thus reconstituted were subjected to Pf3-Lep (amber) and V15D insertion. The insertion activity was determined and is shown at the bottom. In all the autoradiograms, the position of membrane-protected fragment is indicated. Also, the PK-resistant bands unrelated with membrane insertion are indicated by asterisks.

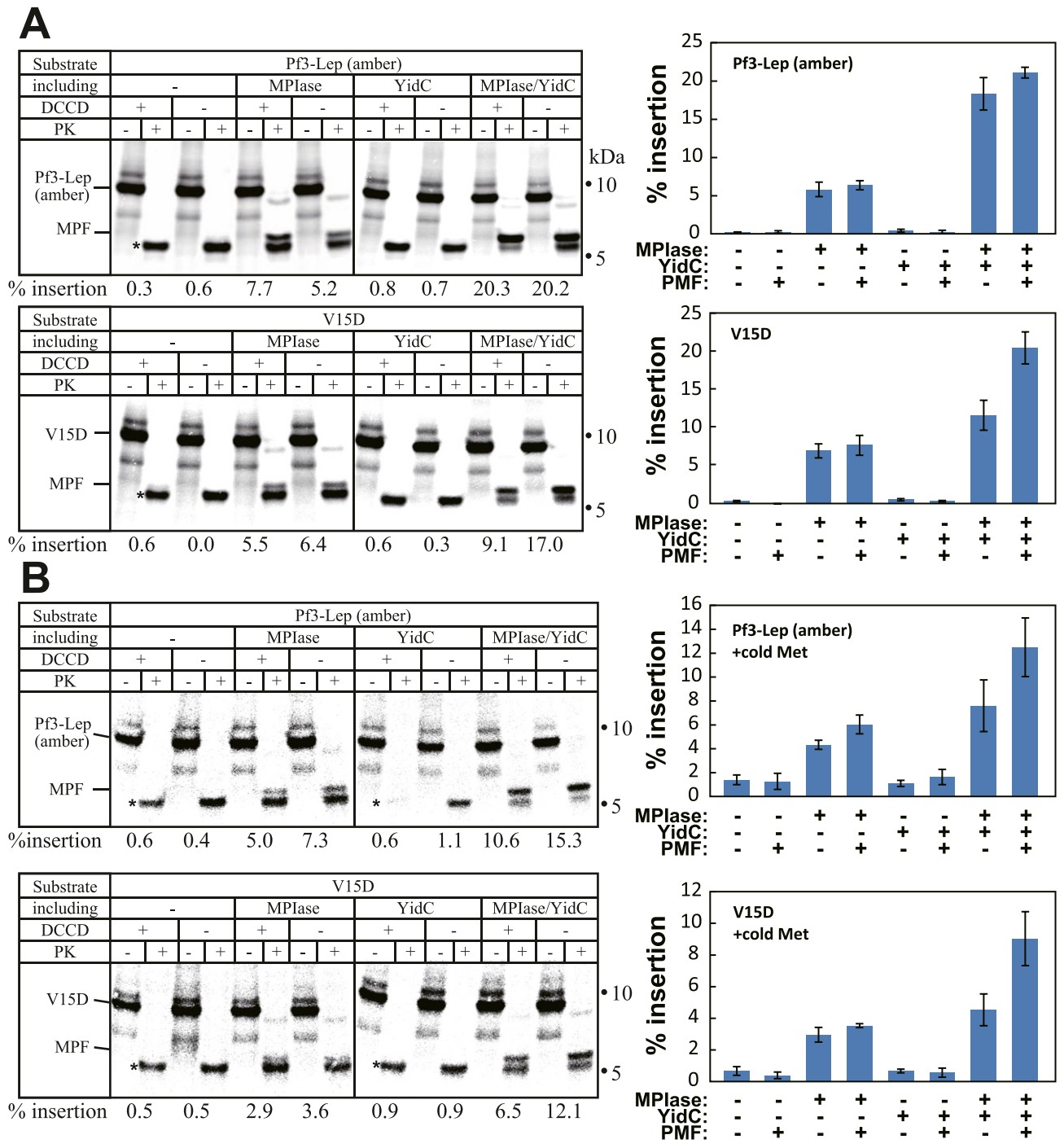

**Figure 4. Expression level of the substrates affects the dependency on insertion factors.**
**(A, B)** Pf3-Lep (amber) and V15D insertion into proteoliposomes becomes dependent on YidC and proton motive force as their expression levels increase. **(A)** The insertion reactions were carried out in the presence of 10 MBq radioactive methionine/ml to increase the expression levels of Pf3-Lep (amber) (upper panel) and V15D (lower panel) (A). **(B)** The expression levels were further increased by adding cold methionine (0.3 mM) (B). The insertion activity for each lot of proteoliposomes was determined as described in the legend to Fig 1D, shown at the bottom. The position of membrane-protected fragment is indicated. The PK-resistant bands unrelated with membrane insertion are indicated by asterisks. The experiments were carried out at least three times. Average activities with error bars are shown at the right of each autoradiogram.

system would be useful to examine the extent of the dependency precisely. Moreover, it is known that the introduction of positively charged residues, such as V4R and V15R, renders insertion SecYEG-dependent in addition (24), strongly suggesting the interplay between SecYEG and MPIase/YidC/PMF. Again, our reconstitution system would be useful to analyze such an interplay. Analysis of the insertion of additional mutants with charge alterations in the N-terminal region is on progress.

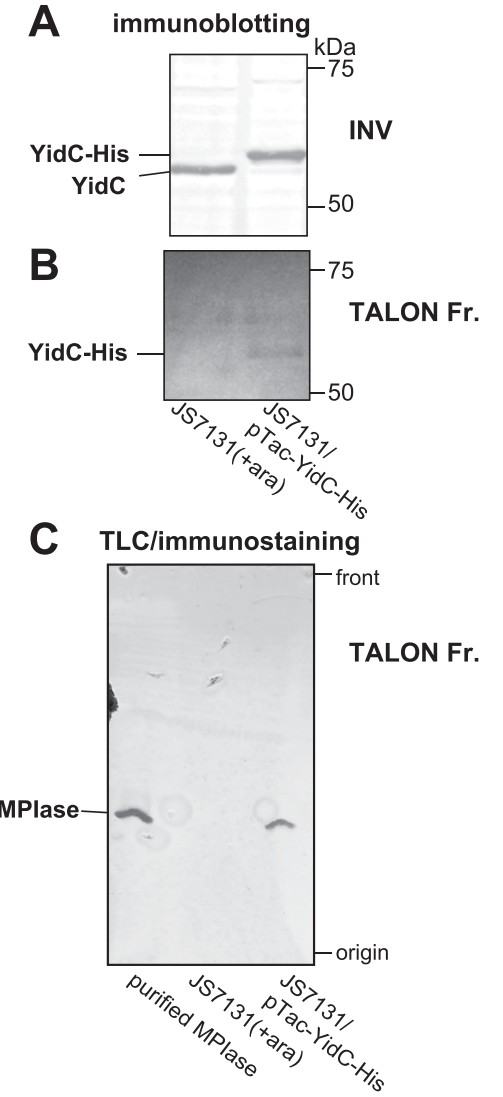

**Figure 5. MPIase and YidC interact directly.**
**(A)** Expression level of YidC in INV (10 μg protein) prepared from JS7131 and JS7131/pTac-YidC-CHis. It was analyzed by immunoblotting using anti-YidC antibody. **(B)** Purification of YidC on TALON column chromatography. Solubilized membranes of each INV were applied onto the TALON column, followed by elution with 150 mM imidazole. The eluted fraction (5 μl) was subjected to immunoblotting using anti-YidC antibody. **(C)** Detection of MPIase in the eluted fractions. **(B)** The eluted fractions in (B) (5 μl) were analyzed on TLC, and then visualized by immunostaining using anti-MPIase antibody. Purified MPIase (10 ng) was used as a standard. The positions of TLC origin/front and MPIase are shown.
Source data are available for this figure.

# Materials and Methods

## Materials

INV were prepared from EK413 (29), KS23 (30), and JS7131 (17) cells, as described (31). To deplete MPIase and YidC, conditionally lethal mutants KS23 and JS7131 were cultivated in the absence of arabinose, respectively. Ffh, FtsY, and YidC were purified from over-producer of the respective proteins as described (7). MPIase was purified from MC4100 as described (15, 16). $F_0F_1$-ATPase from a thermophilic Bacillus PS3 was overproduced in DK8 harboring plasmid pTR19ASDSεΔc (32), a kind gift from Dr. Kuruma (JAMSTEC), and then purified on a TALON column, a cobalt metal-affinity column (Clontech), as described (33). The purified preparation of $F_0F_1$-ATPase was solubilized in 50 mM Hepes/KOH, pH 7.5, 1.5% OG, and 50% (wt/vol) glycerol. Plasmids pLZ1-Pf3-Lep after 61 amber codons and pLZ1-Pf3-Lep V15D after 61 amber codons, which carry genes for Pf3-Lep (amber) and V15D (24) (Fig S1) under the control of the T7 promoter, were provided by Prof Dalbey (Ohio State Univ). Plasmids pT7-7-Pf3 (13) and pIVEX-OmpA (34) were used in vitro to synthesize Pf3 coat protein and OmpA, respectively. Plasmid pTac-YidC-CHis (8) was used to express His-tagged YidC. PL (*E. coli* polar phospholipids) and DAG (dioleoylglycerol) were purchased from Avanti Polar Lipids, Inc. Detergents OG (*n*-octyl-β-D-glucopyranoside) and DDM (*n*-dodecyl β-D-maltoside) were obtained from Dojindo Laboratories. The PURE system, obtained from GeneFrontier Corporation, was optimized for the in vitro inserration assay (15). [35S] EXPRESS Protein Labeling Mix, a mixture containing [35S] methionine and [35S] cysteine (~37 TBq/mmol), was from Perkin Elmer, Inc.

## Reconstitution of (proteo)liposomes

Proteoliposomes were formed by dialysis as follows. PL (500 μg), YidC (40 μg), and $F_0F_1$-ATPase (25 μg), solubilized in 1.5% (wt/vol) OG, were mixed and incubated on ice for 30 min, followed by dialysis against buffer A (50 mM HEPES-KOH, pH 7.5, 1 mM dithiothreitol) for at least 3 h at 4°C. Proteoliposomes, thus reconstituted, were recovered by centrifugation (160,000*g*, 1 h, 4°C) and suspended in buffer A. DAG-containing liposomes were formed by sonication as described (35). PL and DAG (10% amount to PL), mixed in the solvent, were dried under a nitrogen stream and then under vacuum. The dried residues were hydrated in buffer A and allowed to form liposomes through sonication. When necessary, MPIase was mixed with PL and DAG to yield MPIase liposomes. Equal amounts of the YidC/$F_0F_1$-ATPase proteoliposomes and PL/DAG liposomes or PL/DAG/MPIase liposomes were mixed, frozen, thawed and then fused through an extruder with filters of 0.4 μm pore size, as described (7, 13).

## Assaying of protein insertion

The reaction mixture (20 μl), containing the PURE system, SRP (50 μg/ml), FtsY (50 μg/ml), plasmid DNA, [35S] methionine (~2 MBq/ml in Figs 1–3 and ~10 MBq/ml in Fig 4), and INV or proteoliposomes (0.4 mg/ml), was incubated at 37°C for 30 min. Sodium azide (1 mM) or DCCD (0.15 mM) was added when specified. The reaction was terminated by chilling on ice. An aliquot (3 μl) was used to monitor the synthesis level. Another aliquot (15 μl) was treated with 0.5 mg/ml proteinase K (Roche Diagnostics) for 20 min at 25°C. The proteins were precipitated with 5% trichloroacetic acid. After washing the precipitates with acetone, they were analyzed by SDS–PAGE and autoradiography. The radioactive bands were visualized with a Phosphorimager (GE Healthcare) and quantitated using Image-Quant software (GE Healthcare).

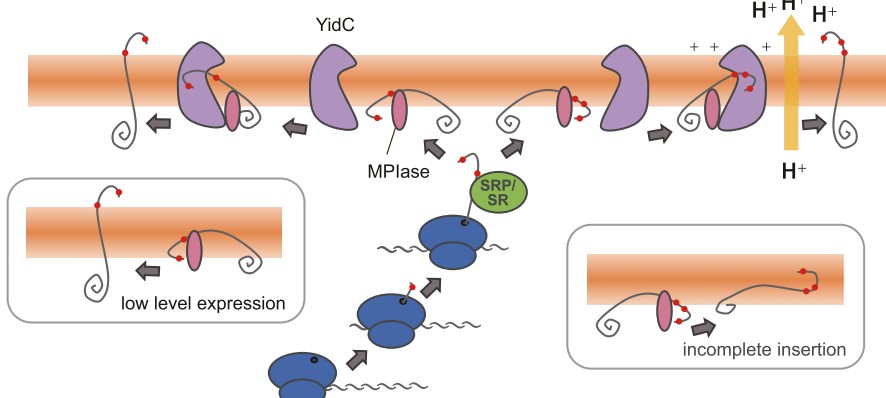

**Figure 6.** **Current model for Pf3-Lep (amber) and V15D insertion into membranes.**
When the expression levels of substrates are low, MPIase is sufficient for insertion (left). As the expression levels of the substrates increase, the MPIase-dependent insertion being to be stimulated by YidC and proton motive force. The yellow arrow represents the generation of proton motive force. The negatively charged residues in the periplasmic region of Pf3-Lep and V15D are denoted by red dots. See the text for details.

### Pull-down assay

INV were prepared from JS7131 cultivated in the presence of 0.2% arabinose or JS7131/pTac-YidC-CHis in the absence of inducers. Both INV (4 mg protein) were treated with 1% DDM, 20% glycerol, 50 mM HEPES-KOH (pH 7.5), followed by recovery of solubilized membranes by centrifugation (170,000$g$, 30 min, 4°C). They were then applied onto the TALON column (0.5 ml). After washing the columns with 20 ml of 0.02% DDM, 10% glycerol, and 50 mM HEPES-KOH (pH 7.5), bound materials were eluted with the same buffer containing 150 mM imidazole. The YidC and MPIase amounts in the eluted fractions were analyzed by immunoblotting by means of anti-YidC antibody and immunostaining of the TLC plates by means of anti-MPIase antibody, respectively. To detect MPIase, the TLC plates were coated before incubation with antibodies, as described (30).

## Supplementary Information

## Acknowledgements

We thank Prof. R. Dalbey, Ohio State University, for the plasmids encoding Pf3-Lep (amber) and the V15D mutant, and for the fruitful discussion; and Ms M Saikudo and Ms M Sawaguchi for the technical assistance. The experiments involving radioisotopes were carried out at the Radio Isotope laboratory of Iwate University. This work was supported by KAKENHI grants (nos. 15KT0073, 17H02209, and 18KK0197 to K Nishiyama, and no. 18J21847 to H Nishikawa). H Nishikawa was a recipient of the Japan Society for the Promotion of Science (JSPS) fellowship.

### Author Contributions

Y Endo: conceptualization, data curation, formal analysis, validation, investigation, visualization, and writing—original draft.
Y Shimizu: investigation and visualization.
H Nishikawa: conceptualization, data curation, formal analysis, funding acquisition, validation, and investigation.
K Sawasato: conceptualization, resources, data curation, and formal analysis.
K Nishiyama: conceptualization, resources, data curation, formal analysis, supervision, funding acquisition, validation, investigation, project administration, and writing—original draft, review, and editing.

### Conflict of Interest Statement

The authors declare that they have no conflict of interest.

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
