## [Reviewer comments · Life Science Alliance]

Life Science Alliance

Interplay between MPlase, YidC and PMF during Sec-independent insertion of membrane proteins

Yuta Endo, Yuko Shimizu, Hanako Nishikawa, Katsuhiro Sawasato, and Ken-ichi Nishiyama
DOI: <https://doi.org/10.26508/lsa.202101162>

Corresponding author(s): Ken-ichi Nishiyama, Iwate University

Review Timeline:	Submission Date:	2021-07-18
	Editorial Decision:	2021-08-11
	Revision Received:	2021-09-28
	Editorial Decision:	2021-09-29
	Revision Received:	2021-09-30
	Accepted:	2021-10-01

Transaction Report:

August 11, 2021

Re: Life Science Alliance manuscript #LSA-2021-01162-T

Prof. Ken-ichi Nishiyama
Iwate University, Faculty of Agriculture
Cryobiofrontier Research Center
3-18-8 Ueda
Morioka 020-8550
Japan

Dear Dr. Nishiyama,

Thank you for submitting your manuscript entitled "Interplay between MPlase, YidC and PMF during Sec-independent insertion of membrane proteins" to Life Science Alliance. The manuscript was assessed by expert reviewers, whose comments are appended to this letter. We invite you to submit a revised manuscript addressing the Reviewer comments. The Bacterial Two-hybrid system experiment is not necessary for further consideration.

Thank you for this interesting contribution to Life Science Alliance. We are looking forward to receiving your revised manuscript.

Sincerely,

- A letter addressing the reviewers' comments point by point.
- An editable version of the final text (.DOC or .DOCX) is needed for copyediting (no PDFs).
- High-resolution figure, supplementary figure and video files uploaded as individual files: See our detailed guidelines for preparing your production-ready images, <https://www.life-science-alliance.org/authors>
- Summary blurb (enter in submission system): A short text summarizing in a single sentence the study (max. 200 characters including spaces). This text is used in conjunction with the titles of papers, hence should be informative and complementary to the title and running title. It should describe the context and significance of the findings for a general readership; it should be written in the present tense and refer to the work in the third person. Author names should not be mentioned.

B. MANUSCRIPT ORGANIZATION AND FORMATTING:

Reviewer #1 (Comments to the Authors (Required)):

Although very limited subset of bacterial proteins can be inserted in apparently unassisted insertion reactions, the vast majority of proteins are not inserted directly and spontaneously into membrane but rather their insertion is guided by protein-conducting channel (Sec translocase) and proteinaceous/non-proteinaceous insertase complex (YidC and MPlase (glycolipid composed of DAG and a glycan chain of three aminosugars linked through pyrophosphate)) with dependence on PMF which can directly (acting electrophoretically) or not (via YidC) contribute to efficiency of insertion process. The dissection of the translocase or insertase assisted and unassisted events in context of PMF requirement is a very important task in this area of research.

The authors of submitted MS used an idealized chimeric protein Pf3-Lep and its derivative V15D mutant with different IN VIVO requirements for insertion factors (i.e inserted in YidC and PMF manner or independently) (J Biol Chem 2013 Mar 15; 288(11): 7704-7716) to dissect IN VITRO their

contribution to insertion efficiency of these model proteins with different net charge of its N-terminal segment (-2 or -3, respectively) Particularly inside-out vesicles (ISOVs) isolated from conditionally lethal YidC and MPlase or proteoliposomes co-reconstituted with YidC, MPlase and F1F0 ATP synthase were utilized to study insertion of vitro synthesized proteins by means of the PURE translation system.

Authors made a great attempt to describe in vitro Sec-independent membrane insertion pathway in which MPlase and YidC and PMF may play an exclusive role.

The major finding can be summarized as follows:

An efficient insertion of both model proteins is observed only when MPlase was present (Fig. 1, E) or co-reconstituted (Fig. 3, B and C) while both YidC and PMF are required for efficient insertion of NT (-3) mutant.

PMF appears to stimulate the membrane translocation of extramembrane segments containing negatively charged residues (EMBO J 1994 May 15;13(10):2267-72 and EMBO J. 1995 Mar 1;14(5):866-75) supporting an existence of electrophoresis-like membrane translocation mechanism as shown on Fig. 5. on the right.

Fig. 2E. If this mechanism exists, I expect that YidC independent yield of insertion should be higher for more net negative N-terminal domain of V15D mutant (-3) however the results are opposite of what I expected in this case. Alternatively, PMF can stimulate an insertion via YidC. Whether these results consistent with the mechanism of PMF- dependent YidC action?

Indeed MPlase-dependent membrane insertion of V15D was not stimulated by YidC in the absence of PMF (Fig. 3B). Although PMF had little effect on MPlase-dependent membrane insertion of V15D, YidC significantly increased insertion yield of V15D in the presence of PMF (Fig. 3C, lane 5 - 8). Reconstitution experiments explain further why DCCD doesn't affect on V15D insertion into ISOVs prepared from YidC-depleted strain. In the absence of PMF, YidC can't stimulate insertion of V15D (Fig.2E, lanes 9,10 vs lanes13, 14).

These results indicate that both YidC and PMF are required for stimulation of membrane insertion of V15D and suggest YidC utilizes PMF to translocate N-terminal domain with net negative charge across the membrane.

I think that authors should emphasize this major finding In Abstract and extend this finding by at least one experiment suggested below.

Although I satisfied with main conclusions however authors should address several essential concerns.

MAJOR

1. Unfortunately, role of MPlase in Sec-dependent and Sec-independent membrane protein insertion is not clarified/explain clearly in Introduction. Whether MPlase is essential factor for both Sec-dependent and Sec-independent membrane protein insertion? Whether SecYEG and YidC are sufficient to induce membrane insertion in vitro? Whether YidC/Sec-independent, YidC only, YidC/PMF only or YidC/Sec mechanisms co-exist and all of them operate in MPlase-dependent manner? The authors should summarize what is known about involvement of glycolipid MPlase in Sec-dependent and Sec-independent membrane protein insertion. Authors should state that MPlase transfer the substrate to SecYEG or YidC, and then SecYEG or YidC complete the

membrane insertion process. Authors can even build a Table for requirement/interdependence of action of these insertion factors.

To show that MPlase is essential for membrane insertion of MtIA, Pf3 coat, 3L-Pf3 coat, M13 procoat, c-subunit of F₀F₁-ATase as shown in Table below. While YidC is not essential for the insertion in vitro, but it significantly stimulates MPlase-dependent membrane insertion. This Table should be extended to PMF requirement.

SecYEG MPlase YidC

MtIA essential essential stimulate

Pf3 coat - essential stimulate

3L-Pf3 coat - essential stimulate

M13 procoat - essential stimulate

F₀c - essential stimulate

2. Fig. 1 and Fig. 5 . It is better to show a net charge of N-terminal domain (either - 2 (two red filled circles) or - 3 (3 red filled circles) for Pf3-Lep and its derivative V15D mutant respectively.

3. Whether the net charge or distribution of negative charges along the translocated acidic N-tail is important for PMF/YidC dependent translocation? Authors can take an advantage of another construct (available from Ross Dalbey' lab) with one positively and one negatively charged residues added to N-terminal segment simultaneously (i.e no net change in charge is made to keep net charge of N-tail of -3) to figure out whether net charge of N-terminal domain or specific positioning /spacing between negatively charged residues is important for the interaction with YidC/PMF. Although positively charged residue added to the N terminal domain leads to insertion via the YidC/Sec pathway, an effect of net negative charge or its positioning on PMF/YidC -dependent insertion can be investigated in proteoliposomes lacking SecYEG.

4. When YidC is depleted, insertion is not affected because it can be handled by the SecYEG or vice versa when SecE is depleted, the protein is inserted by the YidC. If such interchangeability exists how many negative charged residues can YidC handle with PMF assistance? What is the limit for adding of increasing number of negatively charged residues to N-tail translocated by YidC with PMF assistance without presence of SecY protein?

5. Is it possible to rationalize a direct and indirect (via YidC) effect of PMF on translocation of nascent N-terminal domain and summarize in model shown on Fig. 5. and propose as Future Directions?). Whether interaction between Pf3-Lep and its derivative V15D mutant and YidC occurs even in the absence of the PMF? Whether YidC independent mechanism is still possible?

6. Materials and Methods and results shown on Fig. 4.

The author state that "The insertion reactions were carried out in the

presence of 10 MBq radioactive methionine/mL to increase the expression levels of Pf3-Lep (amber) (upper panel) and V15D (lower panel) (A)."

Whether cold methionine was fully omitted in experiments shown on Fig. 1-3 and Fig. 4A?

Is it possible that 5X i.e., much higher specific radioactivity of methionine leads to increase of the detection/sensitivity rather than increase in vitro expression level?

"The expression levels were further increased by adding cold methionine (0.3 mM) (B)".

An excess of cold methionine (0.3 mM) was added to increase a translational loading (Fig. 4). However, a specific activity of [³⁵S] Met was drastically decreased in this case. Whether specific radioactivity was adjusted (increased) accordingly?

MINOR

Abstract

Add "INSERTION" to "... determine the INSERTION factor dependency in vitro" ...

Materials

Authors should state that CONDITIONALLY LETHAL mutants KS23 and JS7131 were utilized to deplete MPlase and YidC, respectively

Reviewer #2 (Comments to the Authors (Required)):

This is an interesting manuscript from Prof. Nishiyama group. The group study how integral membrane proteins are inserted into membranes by YidC and MPlase proteins. The authors have analyzed the effects of YidC and MPlase in the insertion of Pf3-Lep and V15D in the proteoliposome system. Pf3-Lep protein is inserted independently of both YidC and PMF. A variant of Pf3-Lep (V15D) uses both YidC and PMF for its insertion. Accordingly, YidC depletion had no effects on Pf3-Lep insertion but caused reduce V15D insertion in the Proteoliposomes. Both Pf3-Lep and V15D required glycolipid MPlases for insertion into the liposomes.

Q1. The authors suggest that charges in the N-terminal region of Pf3-Lep (variant V15D) and concentration of YidC and MPlase determined the insertion of V15D. I wonder how the interaction between the proteins (YidC and MPlase) with Pf3-Lep is affected by the V15D mutation. The Bacterial Two-hybrid system can be used to test protein-interactions (Pf3-Lep-YidC, Pf3-Lep-MPlase, V15D-YidC and V15D-MPlase). Finding from these studies will enrich this important study.

Q2. A claim of this study is that depending of the use of PL or DAG, Pf3-Lep and V15D can be inserted in the liposomes. In this studies, YidC was not sufficient for insertion of the proteins in

PL+DAG liposomes. While that MPlase was essential for the insertion of both proteins in PL+DAG. These finding may suggest that MPlase and YidC work as a complex and depending of the nature of the liposome both proteins interact and collaborate in the insertion of proteins. I wonder if the authors can immunoprecipitation the complex in native condition. Pull-down experiments can also identified other proteins required for insertion of Pf3-Lep and V15D. This is a nice and elegant study. Great Job!.

Dear Dr. Eric Sawey,

Thank you for the review comments, and we are so happy to have opportunity to revise our manuscript. We have modified text according to the reviewers' comments, together with some additional experiments. Please find our point-by-point response below. We are confident that our revised manuscript is now acceptable for publication in *Life Science Alliance*. We look forward to hearing from you soon.

Best wishes,
Ken-ichi Nishiyama, PhD

Iwate University

Reviewer #1 (Comments to the Authors (Required)):

Although very limited subset of bacterial proteins can be inserted in apparently unassisted insertion reactions, the vast majority of proteins are not inserted directly and spontaneously into membrane but rather their insertion is guided by protein-conducting channel (Sec translocase) and proteinaceous/non-proteinaceous insertase complex (YidC and MPIase (glycolipid composed of DAG and a glycan chain of three aminosugars linked through pyrophosphate)) with dependence on PMF which can directly (acting electrophoretically) or not (via YidC) contribute to efficiency of insertion process. The dissection of the translocase or insertase assisted and unassisted events in context of PMF requirement is a very important task in this area of research.

The authors of submitted MS used an idealized chimeric protein Pf3-Lep and its derivative V15D mutant with different IN VIVO requirements for insertion factors (i.e inserted in YidC and PMF manner or independently) (J Biol Chem 2013 Mar 15; 288(11): 7704-7716) to dissect IN VITRO their contribution to insertion efficiency of these model proteins with different net charge of its N-terminal segment (-2 or -3, respectively) Particularly inside-out vesicles (ISOVs) isolated from conditionally lethal YidC and MPIase or proteoliposomes co-reconstituted with YidC, MPIase and F1F0 ATP synthase were utilized to study insertion of vitro synthesized proteins by means

of the PURE translation system.

Authors made a great attempt to describe in vitro Sec-independent membrane insertion pathway in which MPIase and YidC and PMF may play an exclusive role.

The major finding can be summarized as follows:

An efficient insertion of both model proteins is observed only when MPIase was present (Fig. 1, E) or co-reconstituted (Fig. 3, B and C) while both YidC and PMF are required for efficient insertion of NT (-3) mutant.

PMF appears to stimulate the membrane translocation of extramembrane segments containing negatively charged residues (EMBO J 1994 May 15;13(10):2267-72 and EMBO J. 1995 Mar 1;14(5):866-75) supporting an existence of electrophoresis-like membrane translocation mechanism as shown on Fig. 5. on the right.

Fig. 2E. If this mechanism exists, I expect that YidC independent yield of insertion should be higher for more net negative N-terminal domain of V15D mutant (-3) however the results are opposite of what I expected in this case. Alternatively, PMF can stimulate an insertion via YidC. Whether these results consistent with the mechanism of PMF-dependent YidC action?

Indeed MPIase-dependent membrane insertion of V15D was not stimulated by YidC in the absence of PMF (Fig. 3B). Although PMF had little effect on MPIase-dependent membrane insertion of V15D, YidC significantly increased insertion yield of V15D in the presence of PMF (Fig. 3C, lane 5 - 8).

Reconstitution experiments explain further why DCCD doesn't affect on V15D insertion into ISOVs prepared from YidC-depleted strain. In the absence of PMF, YidC can't stimulate insertion of V15D (Fig.2E, lanes 9,10 vs lanes13, 14).

These results indicate that both YidC and PMF are required for stimulation of membrane insertion of V15D and suggest YidC utilizes PMF to translocate N-terminal domain with net negative charge across the membrane.

I think that authors should emphasize this major finding In Abstract and extend this finding by at least one experiment suggested below.

Although I satisfied with main conclusions however authors should address several

essential concerns.

We are happy to see that this reviewer was satisfied with our conclusions, and appreciate this reviewer's comments.

MAJOR

1. Unfortunately, role of MPIase in Sec-dependent and Sec-independent membrane protein insertion is not clarified/explain clearly in Introduction. Whether MPIase is essential factor for both Sec-dependent and Sec-independent membrane protein insertion? Whether SecYEG and YidC are sufficient to induce membrane insertion in vitro? Whether YidC/Sec-independent, YidC only, YidC/PMF only or YidC/Sec mechanisms co-exist and all of them operate in MPIase-dependent manner? The authors should summarize what is known about involvement of glycolipid MPIase in Sec-dependent and Sec-independent membrane protein insertion. Authors should state that MPIase transfer the substrate to SecYEG or YidC, and then SecYEG or YidC complete the membrane insertion process. Authors can even build a Table for requirement/interdependence of action of these insertion factors.

To show that MPIase is essential for membrane insertion of MtlA, Pf3 coat, 3L-Pf3 coat, M13 procoat, c-subunit of F₀F₁-ATase as shown in Table below. While YidC is not essential for the insertion in vitro, but it significantly stimulates MPIase-dependent membrane insertion. This Table should be extended to PMF requirement.

SecYEG	MPIase	YidC	
MtlA	essential	essential	stimulate
Pf3 coat	essential	stimulate	
3L-Pf3 coat	essential	stimulate	
M13 procoat	essential	stimulate	
F ₀ c	essential	stimulate	

According to the comments, we have modified the Introduction section (P3 L2-4, L17, L21-24, P4 L1-3, L8-9), and added a table (Table 1), summarizing the integration factor dependencies so far determined.

2. Fig. 1 and Fig. 5. It is better to show a net charge of N-terminal domain (either - 2 (two red filled circles) or - 3 (3 red filled circles) for Pf3-Lep and its derivative V15D mutant respectively.

We have modified Figs 1 and 5, showing the net charges.

3. Whether the net charge or distribution of negative charges along the translocated acidic N-tail is important for PMF/YidC dependent translocation? Authors can take an advantage of another construct (available from Ross Dalbey' lab) with one positively and one negatively charged residues added to N-terminal segment simultaneously (i.e no net change in charge is made to keep net charge of N-tail of -3) to figure out whether net charge of N-terminal domain or specific positioning /spacing between negatively charged residues is important for the interaction with YidC/PMF. Although positively charged residue added to the N terminal domain leads to insertion via the YidC/Sec pathway, an effect of net negative charge or its positioning on PMF/YidC -dependent insertion can be investigated in proteoliposomes lacking SecYEG.

We are very much interested in the dependency changes when some positive and negative charges were introduced at the N-terminal region of Pf3-Lep. We have just started to construct a series of mutants to examine the effects of the mutations, and we plan to soon determine the integration activities. However, it should take rather long time to complete. So, we'd like to consider this project as a near future project. In the Discussion section, the statement regarding the changes in dependencies with charge introductions (P10 L14-21).

4. When YidC is depleted, insertion is not affected because it can be handled by the SecYEG or vice versa when SecE is depleted, the protein is inserted by the YidC. If such interchangeability exists how many negative charged residues can YidC handle with PMF assistance? What is the limit for adding of increasing number of negatively charged residues to N-tail translocated by YidC with PMF assistance without

presence of SecY protein?

To answer these questions, a series of mutants should be examined, as well as the comment 3. We have also started to construct such mutants for the project in near future, which was described in the Discussion section (P10 L14-21).

5. Is it possible to rationalize a direct and indirect (via YidC) effect of PMF on translocation of nascent N-terminal domain and summarize in model shown on Fig. 5. and propose as Future Directions?). Whether interaction between Pf3-Lep and its derivative V15D mutant and YidC occurs even in the absence of the PMF? Whether YidC independent mechanism is still possible?

The PMF effect seems significant in the presence of YidC, but not only in the presence of MPIase. Therefore, PMF effect seems to be tightly linked with the YidC function. (P10 L2-3)

When the substrate level increased, YidC stimulated insertion of both proteins even in the absence of PMF, these proteins should interact with YidC. (P9 L18-20)

When the substrate level was low, Pf3-Lep (amber) efficiently inserted with the aid of both MPIase and PMF, independently of YidC. These results are consistent with the *in vivo* results. Therefore, the YidC independent mechanism is still possible. (P10 L8-10)

These were added in the Discussion section, respectively.

6. Materials and Methods and results shown on Fig. 4.

The author state that "The insertion reactions were carried out in the presence of 10 MBq radioactive methionine/mL to increase the expression levels of Pf3-Lep (amber) (upper panel) and V15D (lower panel) (A)."

Whether cold methionine was fully omitted in experiments shown on Fig. 1-3 and Fig. 4A?

Except Fig. 4B, cold methionine was fully omitted.

Is it possible that 5X i.e., much higher specific radioactivity of methionine leads to increase of the detection/sensitivity rather than increase in vitro expression level?

We determined the expression level of Pf3-Lep moieties, by measuring the radioactivities of the bands on autoradiograms. Increase in the amounts of radioactive methionine led not only to increase in detection sensitivity but also increase in the expression level.

“The expression levels were further increased by adding cold methionine (0.3 mM) (B)”.

An excess of cold methionine (0.3 mM) was added to increase a translational loading (Fig. 4). However, a specific activity of [35S] Met was drastically decreased in this case. Whether specific radioactivity was adjusted (increased) accordingly?

As pointed out, the radioactivities of synthesized substrates decreased by adding cold methionine, but the synthesized level was increased significantly. Even if the radioactivities decreased, the synthesized level could be analyzed in a linear range on autoradiograms, while longer exposure time would be necessary.

MINOR

Abstract

Add “INSERTION” to “... determine the INSERTION factor dependency in vitro” ...

Materials

Authors should state that CONDITIONALLY LETHAL mutants KS23 and JS7131 were utilized to deplete MPIase and YidC, respectively

We modified the respective parts (P2 L10, P11 L2), according to the reviewer's comments.

Reviewer #2 (Comments to the Authors (Required)):

This is an interesting manuscript from Prof. Nishiyama group. The group study how integral membrane proteins are inserted into membranes by YidC and MPIase proteins. The authors have analyzed the effects of YidC and MPIase in the insertion of Pf3-Lep and V15D in the proteoliposome system. Pf3-Lep protein is inserted independently of both YidC and PMF. A variant of Pf3-Lep (V15D) uses both YidC and PMF for its insertion. Accordingly, YidC depletion had no effects on Pf3-Lep insertion but caused reduce V15D insertion in the Proteoliposomes. Both Pf3-Lep and V15D required glycolipid MPIase for insertion into the liposomes.

Q1. The authors suggest that charges in the N-terminal region of Pf3-Lep (variant V15D) and concentration of YidC and MPIase determined the insertion of V15D. I wonder how the interaction between the proteins (YidC and MPIase) with Pf3-Lep is affected by the V15D mutation.

The Bacterial Two-hybrid system can be used to test protein-interactions (Pf3-Lep-YidC, Pf3-Lep-MPIase, V15D-YidC and V15D-MPIase). Finding from these studies will enrich this important study.

Since the two-hybrid system between proteins and glycolipids is difficult to perform, we employed the pulldown assay to reveal the direct interaction between YidC and MPIase (see below). Regarding the interaction between the substrates and YidC/MPIase, we are also very interested, however, such interactions would be transient, which makes the detection very difficult. We'd like to analyze such interactions by the SPR analysis in near future.

Q2. A claim of this study is that depending of the use of PL or DAG, Pf3-Lep and V15D can be inserted in the liposomes. In this studies, YidC was not sufficient for

insertion of the proteins in PL+DAG liposomes. While that MPIase was essential for the insertion of both proteins in PL+DAG. These finding may suggest that MPIase and YidC work as a complex and depending of the nature of the liposome both proteins interact and collaborate in the insertion of proteins. I wonder if the authors can immunoprecipitation the complex in native condition. Pull-down experiments can also identified other proteins required for insertion of Pf3-Lep and V15D. This is a nice and elegant study. Great Job!

According to the reviewer's comment, we performed the pull-down assay using the His-tagged YidC. Results (Fig. 5) clearly showed that YidC directly interacts with MPIase, consistently with the interplay between YidC and MPIase. These were added in the text (P8 L13-P9 L2, P9 L16-17, P13 L1-10). We'd like to design the experiments to investigate whether or not substrate membrane proteins affect the interaction in future, because such interaction between substrates and the insertion machinery would be transient.

September 29, 2021

RE: Life Science Alliance Manuscript #LSA-2021-01162-TR

Prof. Ken-ichi Nishiyama
Iwate University
Department of Biological Chemistry and Food Science
3-18-8 Ueda
Morioka 020-8550
Japan

Dear Dr. Nishiyama,

Thank you for submitting your revised manuscript entitled "Interplay between MPlase, YidC and PMF during Sec-independent insertion of membrane proteins". We would be happy to publish your paper in Life Science Alliance pending final revisions necessary to meet our formatting guidelines.

- please add the Twitter handle of your host institute/organization as well as your own or/and one of the authors in our system
- please consult our manuscript preparation guidelines <https://www.life-science-alliance.org/manuscript-prep> and make sure your manuscript sections are in the correct order
- please separate the Results and Discussion section into two - 1. Results 2. Discussion, as per our formatting requirements

Figure checks:

- please add size markers next to each blot
- for the blots in Figure 3C, please don't extend the column bars into the blots themselves (figure 3B is better)

A. FINAL FILES:

B. MANUSCRIPT ORGANIZATION AND FORMATTING:

Sincerely,

October 1, 2021

RE: Life Science Alliance Manuscript #LSA-2021-01162-TRR

Prof. Ken-ichi Nishiyama
Iwate University
Department of Biological Chemistry and Food Science
3-18-8 Ueda
Morioka 020-8550
Japan

Dear Dr. Nishiyama,

Thank you for submitting your Research Article entitled "Interplay between MPlase, YidC and PMF during Sec-independent insertion of membrane proteins". It is a pleasure to let you know that your manuscript is now accepted for publication in Life Science Alliance. Congratulations on this interesting work.

DISTRIBUTION OF MATERIALS:

Again, congratulations on a very nice paper. I hope you found the review process to be constructive and are pleased with how the manuscript was handled editorially. We look forward to future exciting submissions from your lab.

Sincerely,
